# Runoff Simulation under the Effects of the Modified Soil Water Assessment Tool (SWAT) Model in the Jiyun River Basin

Zhaoguang Li [1,2,†], Shan Jian [1,2,†], Rui Gu [1,2] and Jun Sun [2,3,4,*]

1   College of Marine and Environmental Sciences, Tianjin University of Science and Technology, Tianjin 300457, China; g15290419107@163.com (Z.L.); jianshan@tust.edu.cn (S.J.); gurui19990117@163.com (R.G.)
2   Research Centre for Indian Ocean Ecosystem, Tianjin University of Science and Technology, Tianjin 300457, China
3   Institute for Advanced Marine Research, Guangzhou 511462, China
4   State Key Laboratory of Biogeology and Environmental Geology, China University of Geosciences (Wuhan), Wuhan 430074, China
*   Correspondence: phytoplankton@163.com
†   These authors contributed equally to this work.

**Abstract:** Few studies have been conducted to simulate watersheds with insufficient meteorological and hydrological information. The Jiyun River watershed was selected as the study area. A suitable catchment area threshold was determined by combining the river network density method with the Soil and Water Assessment Tool (SWAT) models, which was driven using the CMADS dataset (China Meteorological Assimilation Driving Datasets for the SWAT model). Monthly runoff simulations were conducted for the basin from 2010 to 2014, and the calibration and validation of model parameters were completed with observed data. The results showed that the final expression for the density of the river network in the Jiyun River basin as a function of density ($y$) and the catchment area threshold ($x$) was obtained as $y = 926.782x^{-0.47717}$. The "inflection point" of the exponential function was the optimal catchment area threshold. The catchment area threshold had an upper and lower limit of the applicable range and was related to the percentage of the total basin area. The simulation results would be affected if the threshold values were outside the suitable scope. When the catchment area was 1.42% of the entire watershed area, increasing the threshold value had less effect on the runoff simulation results; decreasing the threshold value would cause the simulation results to be unstable. When the catchment area reached 1.42% to 2.33% of the total watershed area, the simulation results were in good agreement with the observed values; the coefficient of determination ($R^2$) and Nash–Sutcliffe efficiency coefficient ($NSE$) were more significant than 0.79 and 0.78 for the calibration periods evaluation index. Both were greater than 0.77 and 0.76 for the validation period, which met the evaluation requirements of the model. The results showed that the CMADS-driven SWAT model applied to the runoff simulation and the river network density method adoption to determine the catchment area threshold provided a theoretical basis for a reasonable sub-basin division in the Jiyun River basin.

**Keywords:** SWAT model; CMADS; catchment area threshold; runoff simulation

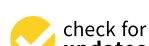



## 1. Introduction

Floods provide a large amount of water resources and cause severe disasters. China is flood-prone and suffers from severe floods annually [1]. The destructive power of floods causes huge losses and significant threats to the safety of people's lives and properties. Various models have been modified and applied to flood simulation, emphasizing flood process simulation [2–4]. Accurate prediction of flood events presupposes the study of runoff processes in different periods, and runoff prediction is of great importance for river flood control [5]. As a hydrological model, the SWAT model can be widely used to study

runoff and water quality simulation of climate change by simulating many physical changes in the water cycle [6]. The SWAT model can be integrated perfectly with a geographic information system (GIS) to provide a runoff simulation [7–15] and a daily timescale for continuous simulations, demonstrating its significant advantages and applicability in flood process simulation. It lays the research foundation for future flood forecasting and provides technical support for flood control.

The application of SWAT models in runoff simulation mainly focuses on studying their applicability and uncertainty analysis. The applicability of model simulation primarily depends on whether the constructed SWAT model can be applied to the research area through relevant evaluation indicators. The uncertainty of the model is mainly related to the accuracy of input data, model parameters, and watershed division accuracy. For the applicability of model simulation, Zhang et al. [16] proved that the SWAT model was suitable for multiple regions, such as the cold northwest part, and high-altitude mountainous regions. Regarding the uncertainty of the model, Meng et al. [17] coupled the snowmelt module with the SWAT model, further improving the accuracy of runoff simulation. Sun et al. [18], by using DEM data of different resolutions to construct SWAT models and other methods, reduced the uncertainty of the model, and the accuracy of runoff simulation was improved. There needs to be more research into the further development of models and inter-model coupling. However, most of the meteorological data and related parameters required by SWAT models were measured by traditional weather stations or calculated. For some watersheds, the scarcity of hydrometeorological data has been a significant problem in hydrological modeling due to the sparse stations, uneven spatial and temporal distribution, or discontinuity of observation time, which essentially restricts the research work. To solve this problem, Fuka [19], and Xu et al. [20] started to apply climate model datasets and reanalysis products to hydrological model simulations. Due to the differences in the fusion methods and data sources used in the generation process of different datasets, the adaptability of different datasets and analysis products in each study area varied [21–23]. Therefore, choosing a climate data analysis product adapted to the study area was essential.

Reanalysis data are developed based on climate models and satellite remote sensing data sources. They have the advantages of uninterrupted regional coverage and high spatial and temporal resolution, making them a potential alternative meteorological data source for hydrological simulations and other applications. Some of the more widely used reanalysis precipitation products include the National Centers for Environmental Prediction (NCEP) Climate Reanalysis Data (CFSR) [24] (1979–2014-07, ~38 km), the National Aeronautics and Space Administration (NASA) Climate Reanalysis Data (MERRA) (1979–2012, ~50 km) [25], the European Centre for Medium-Range Weather Forecasts (ECMWF) (1979–2012, ~82 km), [26] and JRA-55 (1958–2012, ~60 km) from the Japan Meteorological Agency (JMA) [27]. The CMADS (China Meteorological Assimilation Driving Datasets for the SWAT Model, Version 1.1, 2008–2016, ~30 km) reanalysis data covering the East Asia region has a high spatial resolution and provides an option to address the lack of data from meteorological stations in East Asia. The CMADS v1.1 dataset can directly drive the SWAT model, and the CMADS dataset series introduces the China Meteorological Administration's atmospheric assimilation system, using various techniques such as data loop nesting and model extrapolation, with a spatial resolution of $1/4°$, ensuring the density of meteorological data distribution. Meng et al. [28] studied the CMADS, CFSR (Climate Forecast System Reanalysis) dataset, and traditional hydrological station data as examples of meteorological data-driven models for the Heihe River basin. They found that the CMADS series dataset was a better hydrological data-driven model. Qin et al. [29] also showed that the CMADS could be applied to drive the SWAT model in Northeast China. Cao et al. [30] evaluated the accuracy and efficiency of the SWAT model and CMADS for simulating hydrological processes in the fan-shaped Lijiang River basin, China. The results showed that the CMADS performed well in predicting daily streamflow.

Furthermore, the runoff simulation also suffers from the complexity of the terrain and the fact that the river channels and basins extracted based on the original DEM do not match the reality and that the sub-basins are not reflective of the actual conditions within the basin by ArcSWAT itself. The catchment area threshold is the minimum catchment area required to support the permanent existence of a channel. The catchment area threshold directly determines the partitioning of sub-basins within the basin, the sparseness of the river network, and the generation of hydrological response units. As the catchment area threshold varies, the extracted hydrological features vary considerably. More factors are considered for the spatial variation in the basin topography, and the soil elements also change. Since the accuracy of the model prediction results depends to some extent on the description of the relevant characteristics of the watershed by the input spatial variables, the optimal catchment area threshold of the SWAT model can, to some extent, provide a reference for reducing the uncertainty of the simulation results [31]. Xing et al. [32] and Seyler et al. [33] showed that the delineation of the river network was affected by the DEM accuracy. The raw data determine the DEM accuracy and cannot be changed much after the data source is determined. Therefore, the river network can be delineated appropriately by selecting the optimal watershed area threshold to improve the simulation results' accuracy. This paper used the CMADS-driven SWAT model [19] to extract the river network under different catchment area thresholds for reasonably classifying the Jiyun River basin. Firstly, this paper addresses the lack of meteorological data in the study area by coupling the CMADS V1.1 dataset with the SWAT model. A quality assessment of the reliability and applicability of the CMADS dataset in the simulation of runoff from the Jiyun River basin is carried out and the applicability of the CMADS dataset is analyzed. The paper then focuses on the changes in river network density caused by catchment area thresholds and analyzes the effect of subcatchment delineation on model simulation results under different catchment area thresholds. Reasonable catchment area thresholds for the SWAT model are explored. The CMADS-driven SWAT model applied to the runoff simulation to determine the catchment area threshold provided a theoretical basis for a reasonable sub-basin division in the Jiyun River basin and technical support for flood control and resilience.

## 2. Materials and Methods

### 2.1. Study Area

The Jiyun River is an important river in the Haihe basin's northern part. Its source area is situated in Beijing, Hebei Province, Tianjin, and its geographical location is between $116°51'\sim118°15'$ E and $39°46'\sim40°46'$ N (Figure 1). The main tributaries of the Jiyun River are the Juhe and Zhou Rivers. The two branches are called the Jiyun River after they converge in Jiuwangzhuang (Figure 1) Baodi District, Tianjin. The area above Jiuwangzhuang is the main catchment area of the Jiyun River; namely, the upper reaches of the Jiyun River. The Jiyun River flows southeast into the Bohai Sea, with a basin area of 10,288 km$^2$ and a total length of about 189 km, passing through the Binhai New Area in Tianjin. The study area connects the Yanshan Mountains to the North China Plain. The northern and northwestern regions mainly consist of undulating mountains and hills, whereas an area of plains is found in the middle and southern regions. The basin has a large amount of arable land, and the main crops are wheat and corn. The basin is significantly affected by human activities. As a northern, temperate, semi-humid basin with a continental monsoon climate, the study area is subject to precipitation and evaporation of approximately 610–780 and 1225.55 mm/year, respectively. Flooding frequently occurs in summer between June and August, when more than 75% of the annual precipitation occurs. In contrast, the region experiences drier conditions in the winter, with only 2% of the rainfall occurring from December to February.

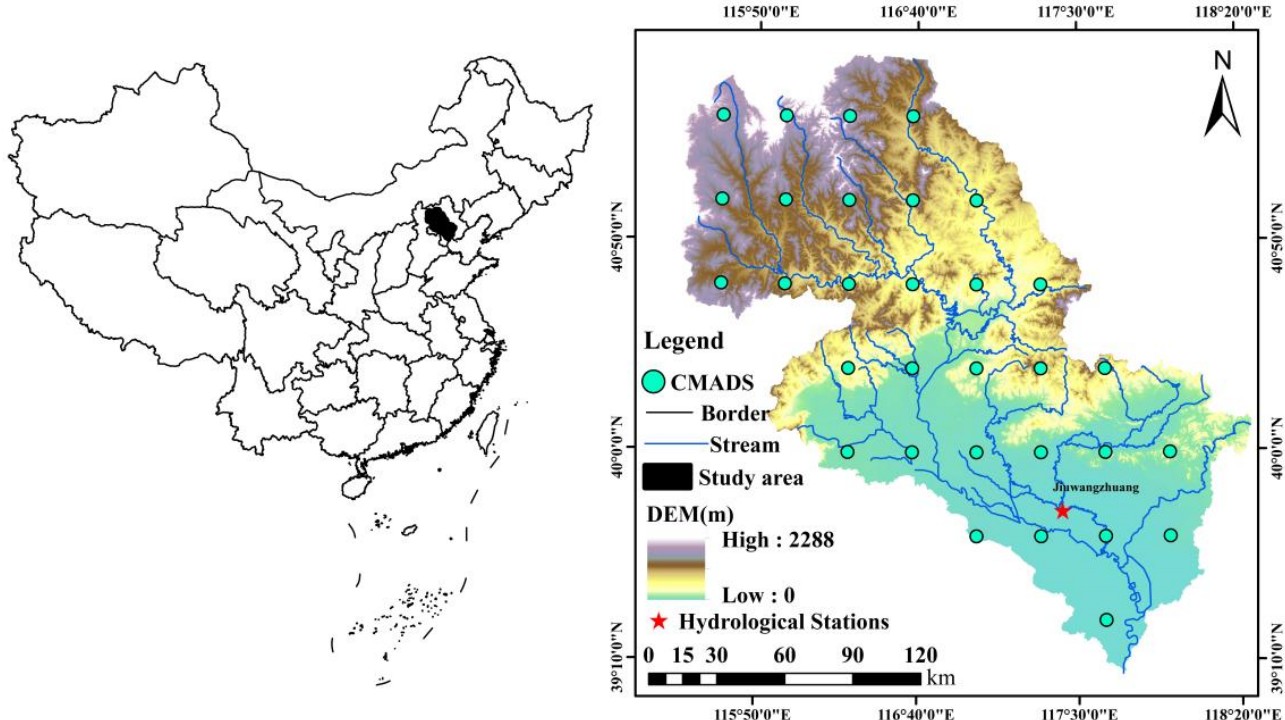

**Figure 1.** Location of the study area and distribution of the hydrological and meteorological gauges.

*2.2. Data Analysis*

The SWAT model study's data are divided into spatial and attribute data. Spatial geographic information data include Digital Elevation Model (DEM), land use/land cover change (LUCC), and soil data. The attribute data are meteorological data, soil attribute data, and hydrological station runoff data. The DEM, land use, soil, and other spatial geographic information data are all in the Beijing 1954 coordinate system and the CGCS2000 3 Degree GK CK 117E projection system. The data sources are shown in Table 1.

**Table 1.** Data description for the study area.

| Data | Source | Resolution |
| --- | --- | --- |
| DEM | ASTER GDEM https://earthexplorer.usgs.gov/ (accessed on 2 March 2021) | 30 m |
| Land use | Landsat-8 https://earthexplorer.usgs.gov/ (accessed on 5 March 2021) | 30 m |
| Soil | HWSD https://www.fao.org/ (accessed on 7 March 2021) | 30 m |
| Weather | CMADS version 1.1 http://www.cmads.org/ (accessed on 8 March 2021) | 28 km |

(1) The DEM data contain natural depressions and pseudo-depressions caused by data errors, which makes it impossible to determine the specific direction of water flow [34–37]. Using these data directly will result in a discontinuous river network [31]. Hence, the data need to be preprocessed to fill the depressions. The topography and geomorphology of the Jiyun River basin are complex, the elevation information cannot be expressed accurately by DEM, and the D8 algorithm within the SWAT model cannot accurately calculate the river network location to define the river flow direction [31,38]. The "burn-in" algorithm converts the digital river systems into raster data of the same size as the DEM raster cells, then reduces the elevation values of the DEM river, and ensures that the corrected channel elevation values are lower than the elevation values of the surrounding areas of the channels so that the final extracted drains are closer to the actual river network.

(2) Land use and soil properties affect different hydrological units' flow production and sink characteristics [39,40]. After reclassification, the land was divided into six primary types: arable land, forest, grassland, water area, unused land, and urban and rural indus-

trial, mining, and residential land. Of these, arable land occupies 57.24% of the total area; forest 16.40%; grassland 6.45%; watershed 3.96%; unused land 0.11%; and urban and rural industrial, mining, and residential land 12.93%. Arable land is the most significant land type, occupying the majority of land, followed by forest and urban and rural industrial, mining, and residential land (Figure 2). This shows that the land use types in the Jiyun River predominantly represent land under human influence.

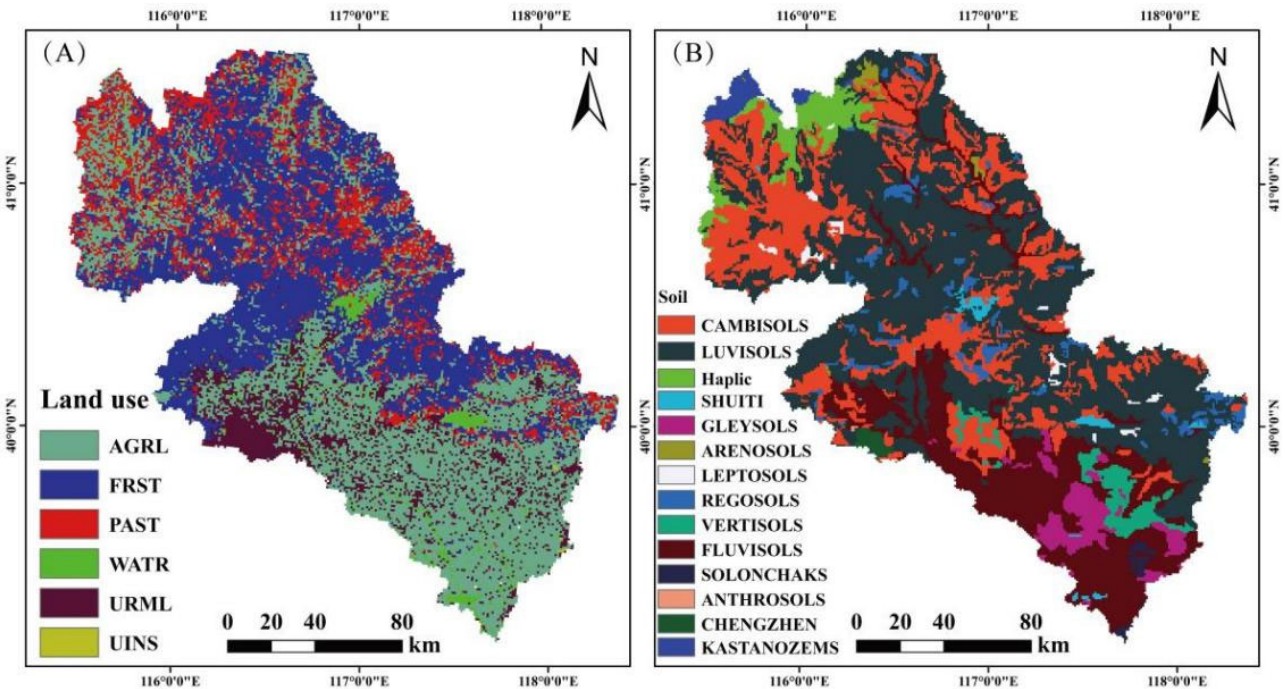

**Figure 2.** Distribution of soil types (**A**) and land cover types (**B**).

The impact of land use change on runoff is mainly reflected in the redistribution of precipitation, and land use change is also a measure for soil and water conservation. On the one hand, land use change affects precipitation and runoff by influencing the vegetation cover of the watershed, thus changing the production and sink mechanisms and hydrological processes; on the other hand, land use change increases the impervious surface area and affects the infiltration capacity of the soil, resulting in a decrease in subsurface runoff. Land use and land cover changes alter the topography of the study area, affecting not only the regional hydrological cycle but also the regional hydrological and ecological processes in a more obvious way.

(3) The soils were classified into 11 major categories: fishpond, vertisols, solonchaks, water bodies, regosols, luvisols, rankers, gleysols, fluvisols, cambisols, and arenosols. The percentages of the different soils in the watershed area are fishpond 0.02%; vertisols 5.92%; solonchaks 1.07%; water bodies 1.06%; regosols 3.17%; luvisols 33.57%; rankers 0.98%; gleysols 8.32%; fluvisols 34.39%; cambisols 11.45%; and arenosols 0.05%. Luvisols and fluvisols are the main soil types (Figure 2). A soil database was established, and soil hydrology was grouped according to the minimum infiltration rate of the soil parameters. SPAW software calculated soil parameters such as saturated hydraulic conductivity, wet soil capacity, and available soil water in the study area. The anion-exchange porosity, potential fracture volume of the soil profile, and surface reflectance rate were taken as 0.5, 0.5, and 0.01, respectively, following the Chinese soil structure. Soil erosion factors (USLE-K) were calculated concerning the modified EPIC model [41]. We referred to the World Soil Database for the organic carbon content of clay and electrical conductivity. Details are given in Table 2.

**Table 2.** Properties of soil physical parameters in the Jiyun River basin.

| Soil Type | Model Code | Proportion | SOL_K | SOL_BD | SOL_AWC | SOL_CBN | CLAY | SILT | SAND | ROCK | USLE-K |
|---|---|---|---|---|---|---|---|---|---|---|---|
| Rudiment soil | CAMBISOLS | 11.45% | 8.79 | 1.52 | 0.14 | 0.65 | 21 | 43 | 36 | 6 | 0.33 |
| Highly active leached soil | LUVISOLS | 33.57% | 60.07 | 1.55 | 0.06 | 0.63 | 9 | 11 | 80 | 4 | 0.17 |
| Alluvial soil | FLUVISOLS | 34.39% | 9.32 | 1.53 | 0.13 | 0.6 | 18 | 48 | 34 | 15 | 0.35 |
| Gley soil | GLEYSOLS | 8.32% | 15.29 | 1.46 | 0.14 | 1.3 | 19 | 40 | 41 | 4 | 0.27 |
| Sandy soil | ARENOSOLS | 0.05% | 9.91 | 1.6 | 0.08 | 0.43 | 24 | 10 | 66 | 10 | 0.18 |
| Thin layer soil | LEPTOSOLS | 0.98% | 18.35 | 1.33 | 0.14 | 2.13 | 19 | 44 | 37 | 16 | 0.26 |
| Loose lithologic soil | REGOSOLS | 3.17% | 9.95 | 1.52 | 0.12 | 0.75 | 21 | 35 | 44 | 14 | 0.29 |
| Denatured soil | VERTISOLS | 5.92% | 6.76 | 1.44 | 0.12 | 1.5 | 29 | 27 | 44 | 7 | 0.22 |
| Saline soil | SOLONCHAKS | 1.07% | 5.62 | 1.52 | 0.13 | 0.46 | 25 | 41 | 34 | 7 | 0.33 |
| Water body | SHUITI | 1.06% | 5.62 | 1.52 | 0.13 | 0.46 | 25 | 41 | 34 | 7 | 0.33 |
| Fish pond | YUTANG | 0.02% | 15.29 | 1.46 | 0.14 | 1.3 | 19 | 40 | 41 | 4 | 0.27 |

Notes: SOL_K: saturated hydraulic conductivity (mm/h); SOL_BD: soil wet capacity (g/cm$^3$); SOL_AWC: soil available water (mm); SOL_CBN: organic carbon content; CLAY: clay; SILT: chalk; SAND: sand; ROCK: gravel.

Soil hydrological properties influence hydrological processes, climate change, and carbon and nitrogen cycle processes. The spatial distribution of soil physical properties strongly influences soil moisture, which determines processes such as groundwater recharge and unsaturated soil evaporation. The infiltration process at the profile level is the link between surface water, groundwater, and soil water distribution, and the mechanism by which soil hydrological properties influence the infiltration process is also an essential link to hydrological processes such as rainfall runoff. The main hydrological process at the slope scale is the slope surface flow production process, where the critical soil hydrological properties are topography, soil hydraulic properties, soil porosity, and soil layer thickness. The vital soil hydrological properties are soil saturation, hydraulic conductivity, soil texture, stratigraphic characteristics, and geological conditions. Analyzing soil hydrological properties can provide theoretical support for studying hydrological processes and mechanisms at the regional scale.

(4) In China, few large-scale meteorological datasets cover rainfall, temperature, relative humidity, wind speed, and solar radiation. Traditional meteorological observation stations are scarce and unevenly distributed. As a result, Dr. Xianyong Meng from China Agricultural University used Spatio-Temporal Multiscale Analysis System (STMAS) assimilation and big data techniques to calibrate nearly 10,000 regional automatic stations of the European Centre for Medium-Range Weather Forecasting (ECMWF) in China to establish the CMADS dataset, to make it reflect the ground-based meteorological conditions more closely [28,42]. CMADS made appropriate adjustments to the different formats of the SWAT input files so that SWAT models can be input directly without format changes to, for example, daily rainfall, maximum and minimum temperatures, mean wind speed, relative humidity, and solar radiation meteorological data. The SWAT model can thus be run directly without changing the data type [16,30].

(5) Monthly flow data for the hydrological stations were obtained from the Tianjin Flood Control and Drought Relief Headquarters.

### 2.3. Hydrological Modeling

The SWAT hydrologic model simulates hydrologic processes, categorized as land surface and riverine processes. Land surface processes are the internal circulation and transformation of runoff, sediment, and nitrogen and phosphorus pollutants in river sub-basins, which converge with the main river channels through each sub-basin. The riverine process simulates the transport of river runoff, sediment, and nitrogen and phosphorus pollutants through the river sub-basins to the outlet of the river basin. Its water balance equation is:

$$SW_{t,i} = SW_{0,i} + \sum_{i=1}^{t} \left( R_{day,i} - Q_{surf,i} - E_{a,i} - W_{seep,i} - Q_{gw,i} \right) \tag{1}$$

where $SW_{t,i}$ is final soil water content (mm); $SW_{0,i}$ is the initial water content (mm); $t$ is time step (day); $R_{day,i}$, $Q_{surf,i}$, and $E_{a,i}$ are the amounts of precipitation, surface runoff, and

evaporation on day $i$ (mm), respectively; $W_{seep,i}$ is the amount of water entering the vadose zone from soil profile on day $i$ (mm), and $Q_{gw,i}$ is the amount of the return flows on day $i$ (mm) [43].

### 2.4. River Network Density Method

River network density refers to the ratio of the total length of main and tributary streams in a watershed to the basin area or the whole length of the natural and artificial streams in a unit area. The river network density method calculates the optimal catchment area threshold by constructing a correlation between the river network density and the threshold value. As the threshold value increases, the number of grids classified as river channels decreases. The number of class 1 and 2 rivers is reduced after vectorization, and the river network density is affected by this and decreases in proportion. By calculating the river network density under different threshold values, the relationship curve between river network density and the threshold value is obtained, and the threshold value corresponding to the smooth change region of the curve is taken as the optimal reasonable threshold value. The calculation formula for river network density is

$$D = L/S \tag{2}$$

where $D$ is the river network density, $L$ is the length of the river under the corresponding threshold, and $S$ is the area of the study area.

ArcGIS generates the total length of waterways under different thresholds, counts the length of rivers, and calculates the river network density. The image of river network density and point is drawn by fitting the functional relationship. Finally, the coordinates where the river network density tends to be smooth with the change in threshold are found according to the location of the inflection point. Better accuracy can be obtained by using the entry of this point for river network extraction.

### 2.5. Model Setup

Information was extracted from the digital elevation model map to divide the study area into several sub-basins for applying the SWAT model. Each sub-basin was redivided into corresponding hydrological response units by overlaying land use and soil spatial data and setting the slope. Based on the D8 algorithm and the "burn-in" algorithm, this study combined the steepest slope principle and the concept of minimum catchment area threshold to automatically generate the river network of a watershed and the topological relationships between sub-basins. The smaller the threshold setting, the more detailed the river network and the smaller the sub-basin area. In the simulation process of the SWAT model, three time periods of 2008–2009, 2010–2011, and 2012–2013 were set as the model warm-up period, the calibration period, and the verification period, respectively, in the Jiyun River basin.

### 2.6. Calibration and Validation

To evaluate the simulation accuracy of the model and whether the calibration results were satisfactory, the Nash–Sutcliffe efficiency coefficient (*NSE*), the coefficient of determination ($R^2$), and the percent bias (*PBIAS*) were chosen as the quantitative evaluation indices of the applicability of the model. *NSE* is a standard statistical equation that reflects the degree of fit between the observed values and the corresponding simulated values. The *NSE* is calculated in the range of $-\infty \sim 1$. When *NSE* = 1, it can be considered that the simulated value fits perfectly with the observed value; when $0.5 < NSE < 1$, the simulation result of the model is acceptable, and when $NSE < 0$, it can be considered that the model simulation result is poor. The coefficient of determination, $R^2$, characterizes the correlation between the measured variables, referring to the relationship between measured and simulated. *PBIAS* is a vital evaluation index for the efficiency of hydrological model simulations, which quantifies the water balance error and measures the model's performance by comparing the average trend of observed and simulated values. The optimal value of *PBIAS* is 0, and the

model simulation results are acceptable when $|PBIAS| < 25\%$. Santhi [44] suggested that the model simulation results can be obtained by considering $R^2 > 0.5$ and $NSE > 0.5$ as good SWAT model simulation criteria. Ahmad [45] argued that $R^2 > 0.5$ and $NSE > 0.4$ could also be used as a criterion for good model metrics. Moriasi [46] thought that the model simulation could be judged as good if $NSE > 0.50$ and $R^2 \leq 0.7$, and if $|PBIAS| \leq 25\%$ for streamflow, $|PBIAS| \leq 55\%$ for sediment, and $|PBIAS| \leq 75\%$ for N and P. If the metrics are met, the model can be considered suitable for simulation flow-through processes in the basin. Concerning the above evaluation system for simulation results, this was classified into four grades. The grading is shown in Table 3. The detailed equations for the evaluation indicators are as follows.

**Table 3.** Evaluation of SWAT model simulation results.

| Result Grade | *NSE* | $R^2$ | *PBIAS* (%) |
|---|---|---|---|
| Very good | $0.75 < NSE \leq 1.00$ | $0.80 < R^2 \leq 1.00$ | $PBIAS < \pm 10$ |
| Good | $0.65 < NSE \leq 0.75$ | $0.70 < R^2 \leq 0.80$ | $\pm 10 \leq PBIAS < \pm 15$ |
| Satisfactory | $0.50 < NSE \leq 0.65$ | $0.50 < R^2 \leq 0.70$ | $\pm 15 \leq PBIAS < \pm 25$ |
| Unsatisfactory | $NSE \leq 0.50$ | $R^2 \leq 0.50$ | $PBIAS \geq \pm 25$ |

The Nash–Sutcliffe efficiency coefficient (*NSE*) equation is

$$NSE = 1 - \frac{\sum_{i=1}^{n}(Q_{obs} - Q_{sim})^2}{\sum_{i=1}^{n}(Q_{obs} - \overline{Q_{obs}})^2} \tag{3}$$

The coefficient of determination ($R^2$) is obtained using

$$R^2 = \frac{\left[\sum_{i=1}^{n}(Q_{obs} - Q_{avg}) \times (Q_{sim} - \overline{Q_{sim}})\right]^2}{\sum_{i=1}^{n}(Q_{obs} - Q_{avg})^2 \times \sum_{i=1}^{n}(Q_{sim} - \overline{Q_{sim}})^2} \tag{4}$$

The percent bias (*PBIAS*) formula is

$$PBIAS = \frac{\sum_{i=1}^{n}(Q_{obs} - Q_{sim})}{\sum_{i=1}^{n} Q_{obs}} \times 100 \tag{5}$$

In the above equation, $Q_{obs}$ is the actual observed value of the data; $Q_{sim}$ is the simulated value of the data; $Q_{avg}$ is the observed mean of the data; $\overline{Q_{sim}}$ is the simulated mean of the data; and $n$ is the number of samples.

*2.7. Sensitivity Analysis*

The parameter sensitivity test was an essential step in evaluating the results that affect the calibration and verification of the model. It would take a lot of time and effort to adjust each parameter one by one manually, and the value of the manual calibration is also easily affected by human subjective factors. SWAT-CUP is software for calibrating SWAT models, which can be used to perform calibration, verification, sensitivity, and uncertainty analysis.

This study used SWAT-CUP software for parameter sensitivity analysis and calibration [47,48]. The SUF1-2 algorithm in SWAT-CUP was used, whereas the parameters were selected using the Latin hyper-dimension method [49–52], which was calculated by random sampling. Then, the selected sensitivity parameters and parameter value ranges were obtained. The $t$ and $p$ values evaluate the sensitivity and identify parameters sensitive to runoff processes. The larger the absolute value of $t$, the greater the sensitivity of the parameter; the $p$-value reflects the significance of the sensitivity, and the closer the $p$-value is to 0, the more significant the sensitivity of the parameter. The higher the absolute value of t, the greater the sensitivity of the parameter [52,53].

SWAT-CUP was used for sensitivity analysis of the model to filter out the parameters with little influence and improve the efficiency of the model calibration determination. The

runoff-related SWAT model parameters were chosen using sensitivity analysis. From this, nine parameters significantly influencing the model were selected for model calibration determination, and their sensitivity ranking is shown in Table 4.

**Table 4.** Parameter sensitivity of SWAT model.

| Susceptibility | Parameter | Definition | $t$-Value | $p$-Value | Range |
|:---:|:---:|:---:|:---:|:---:|:---:|
| 1 | SOL_K | Saturated hydraulic conductivity of first layer (mm/h) | 18.75 | 0.0000001 | −1.2 |
| 2 | SOL_BD | Moist bulk density (g/cm$^3$) | 10.65 | 0.0000001 | −1.2 |
| 3 | SOL_Z | Depth from soil surface to bottom of layer (mm) | −9.27 | 0.0076368 | 10–3500 |
| 4 | RCHRG_DP | Deep aquifer percolation fraction | 8.99 | 0.0253185 | 0–1 |
| 5 | REVAPMN | Depth of water for evaporation (mm) | 4.32 | 0.1000428 | 0–500 |
| 6 | ALPHA_BNK | Recession constant value of base flow | −3.76 | 0.1079156 | 0–1 |
| 7 | GW_REVAP | Groundwater evaporation coefficient | −2.27 | 0.1080458 | 0.02–0.2 |
| 8 | CANMX | Maximum canopy storage (mm) | −1.56 | 0.3081987 | 0–10 |
| 9 | HRU_SLP | Average slope steepness (m/m) | 1.33 | 0.31099899 | 0–1 |

The physical soil traits control water and air movement across the soil slope and significantly impact water circulation within the HRU. Inputs for chemical characteristics were used to set initial values for the different chemicals in the soil. RCHRG_DP has essential roles in hydrologic simulation. SOL_K relates the flow rate (flux density) of soil water to the hydraulic gradient and measures the ease of water movement in the soil. SOL_BD is the ratio of the mass of solid particles to the total volume of the soil and reflects the degree of soil density. SOL_Z measures soil water. RCHRG_DP is an indicator of the comprehensive infiltration capacity of the soil. CANMX is the maximum canopy retention, which shows the significant effect of crop canopy on infiltration, surface runoff, and evaporation. GW_REVAP is the submersible recharge coefficient, which refers to the infiltration capacity of groundwater in the soil and is a critical fluid flow parameter. REVAPMN is the shallow groundwater re-evaporation coefficient, which indicates the evaporation depth threshold of shallow aquifers. ALPHA_BNK is the baseflow recession constant, which affects water flow in the main river or river within a sub-basin. HRU_SLP is the average slope of the sub-basin. In hydrology, its most direct effect is the flow velocity, bringing about more hydrodynamically induced energy transfers, such as the conversion of potential and kinetic energy. As can be seen from the table, the study area simulation was particularly sensitive to the soil database parameters. The four most sensitive parameters for runoff simulation in the study area were saturated soil hydraulic conductivity (SOL_K), soil bulk density (SOL_BD), bottom soil depth (SOL_Z), and deep storage infiltration coefficient (RCHRG_DP).

### 3. Results and Discussion

*3.1. Sub-Basin Division*

The river network density method was used to select the best threshold conditions for the river network division. The catchment areas were set as 17, 27, 37, 47, 57, 67, 77, 87, 97, 107, 117, and 147 km$^2$. Finally, 12 river network division schemes were obtained. The different sub-basin classification schemes are shown in Figure 3.

The catchment area threshold and river network density were presented as scatterplots using five function analyses: linear, exponential, logarithmic, polynomial, and power function for trend line fitting. The power function was found to be the best fit. Using the river network density method, we took the second derivative of the power function and performed trend fitting again (Figure 4). According to the river network density method, it was known that the catchment threshold area was best when the river network density tended to flatten out with the threshold change. The functional density expression was $y = 926.782x^{-0.47717}$. As the catchment area threshold increases, the river network density approaches infinitely close to 0. It took a lot of work to solve the turning point to extract the catchment threshold using the first-order and second-order derivatives of the power

function. Therefore, this paper proposes to extract the catchment area threshold by the tangent of the power function to a straight line.

$$y = 926.782x^{-0.47717} \tag{6}$$

$$y = -x + a \tag{7}$$

where $y$ is the river network density in $km^{-1}$, $x$ is the catchment area in $km^2$, and a is the intercept.

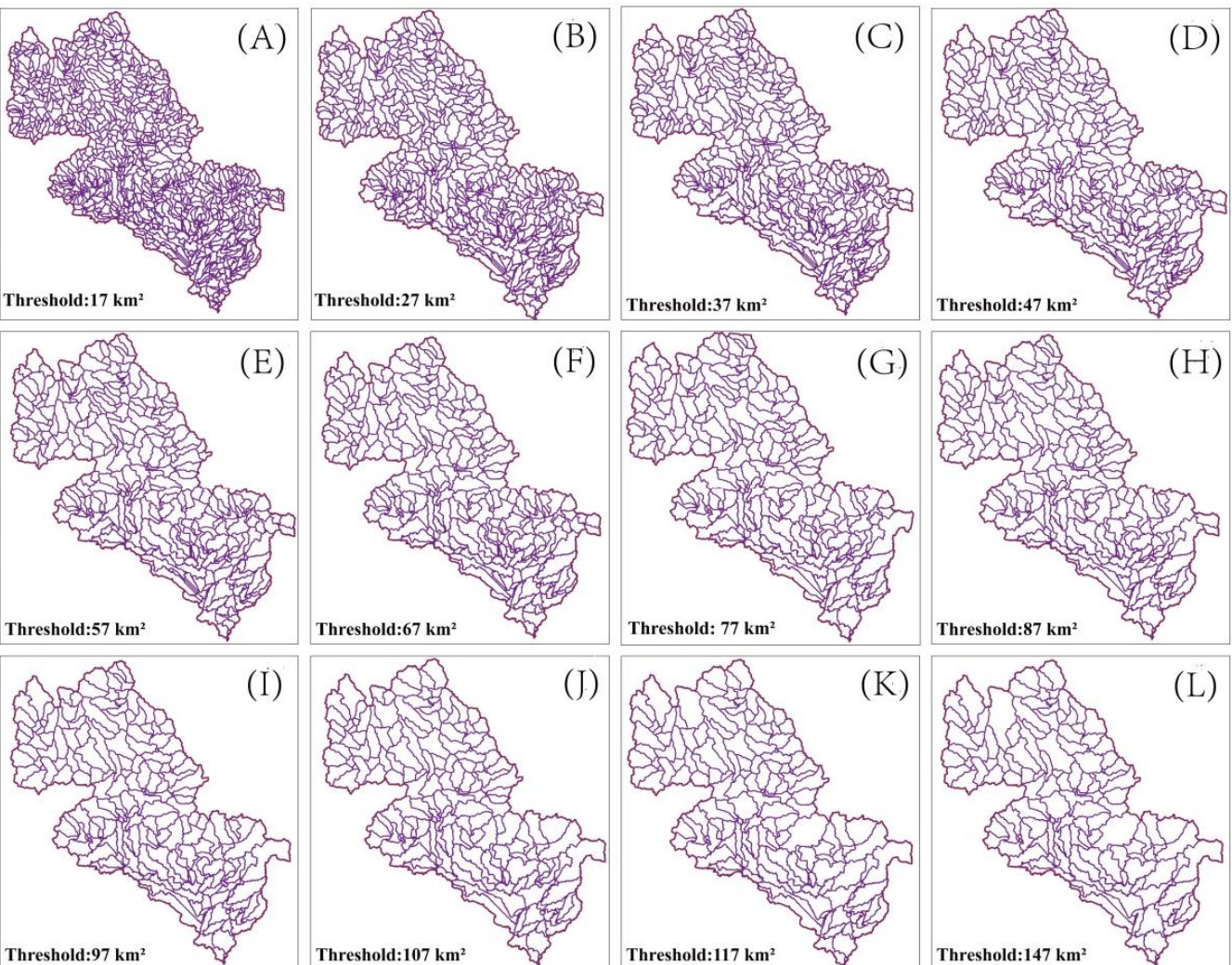

**Figure 3.** Comparison of subwatershed divisions with different catchment area thresholds ((**A**–**L**) represents subwatershed division under the catchment area threshold 17–147).

The point where the power function is tangent to the equation of the line is the inflection point, also called the stability point in mathematics. Before the point of stability, the rate of change in catchment area threshold $\Delta x$ is smaller than the rate of change in river network density $\Delta y$, and vice versa $\Delta y < \Delta x$. The inflection point is found when the density of the river network is stable with the change in the catchment area threshold, and the reasonable catchment area threshold is obtained. When $x$ has a unique solution, $\Delta y = \Delta x$, the value taken is the optimal catchment area threshold. After calculation, when a = 191.33037, the unknown $x$ of the system of equations has a unique solution of 61.80518, and thus the optimal threshold area of the study area is determined to be 61.80518.

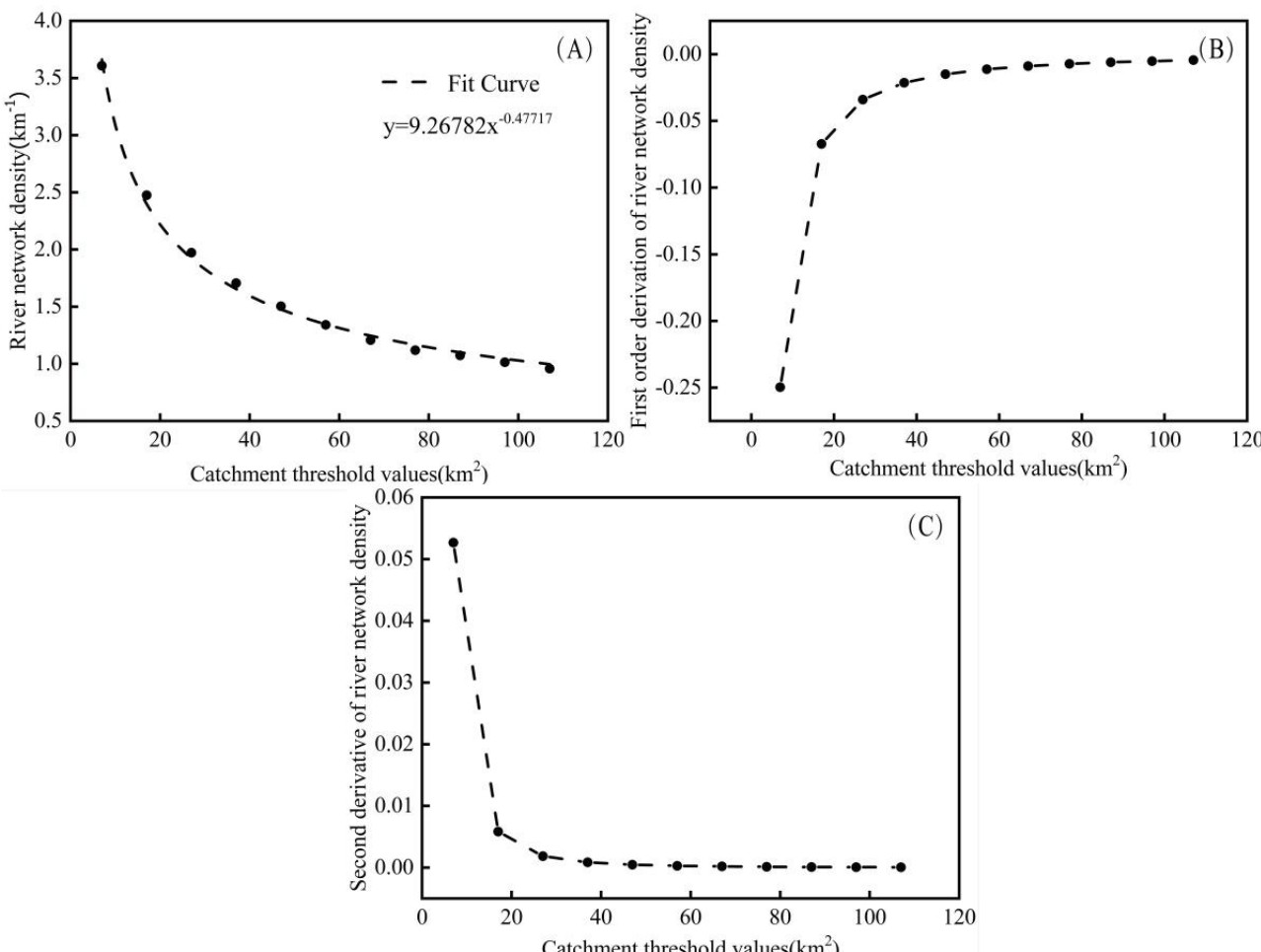

**Figure 4.** Variation in river network density (**A**), first−order derivatives of river network density (**B**), and second−order derivatives of river network density (**C**) with catchment thresholds.

As the catchment area increases, the starting point of the rivers on the slope will move closer to the adjacent flat terrain area in the watershed. The number of rivers on the slope decreases, and the pseudo rivers are removed. The river network density and catchment threshold power function are coupled with the tangent equation. A numerical method with the rate of change in river network density equal to the rate of change in catchment value is adopted to determine the optimal catchment threshold in the region. Better accuracy can be obtained by extracting the river network with the threshold value at this point. This method can effectively avoid the influence of human subjectivity.

*3.2. Runoff Simulation*

It can be seen that the variation trend of the simulated value of the flood was consistent with the measured value, whereas the peaks and valleys of runoff as a whole were better simulated (Figure 5A,B). The model can reasonably simulate the runoff from June to October. The flood periods of the measured and simulated flows are mainly concentrated in June, July, and August, consistent with the study area's perennial meteorological rainfall conditions. The most significant runoff was in August and the second largest in July. The runoff was smaller in winter. The maximum monthly runoff in the calibration period on August 2011 was 12.42 $m^3 \cdot s^{-1}$, whereas the simulated monthly runoff was 12.84 $m^3 \cdot s^{-1}$. The measured runoff in flood season was 0.42 $m^3 \cdot s^{-1}$ lower than the simulated runoff. On August 2012, the measured runoff reached the highest value of 27.45 $m^3 \cdot s^{-1}$, whereas the simulated runoff was 32.98 $m^3 \cdot s^{-1}$. The measured runoff in flood season was 5.53 $m^3 \cdot s^{-1}$ higher than the simulated runoff.

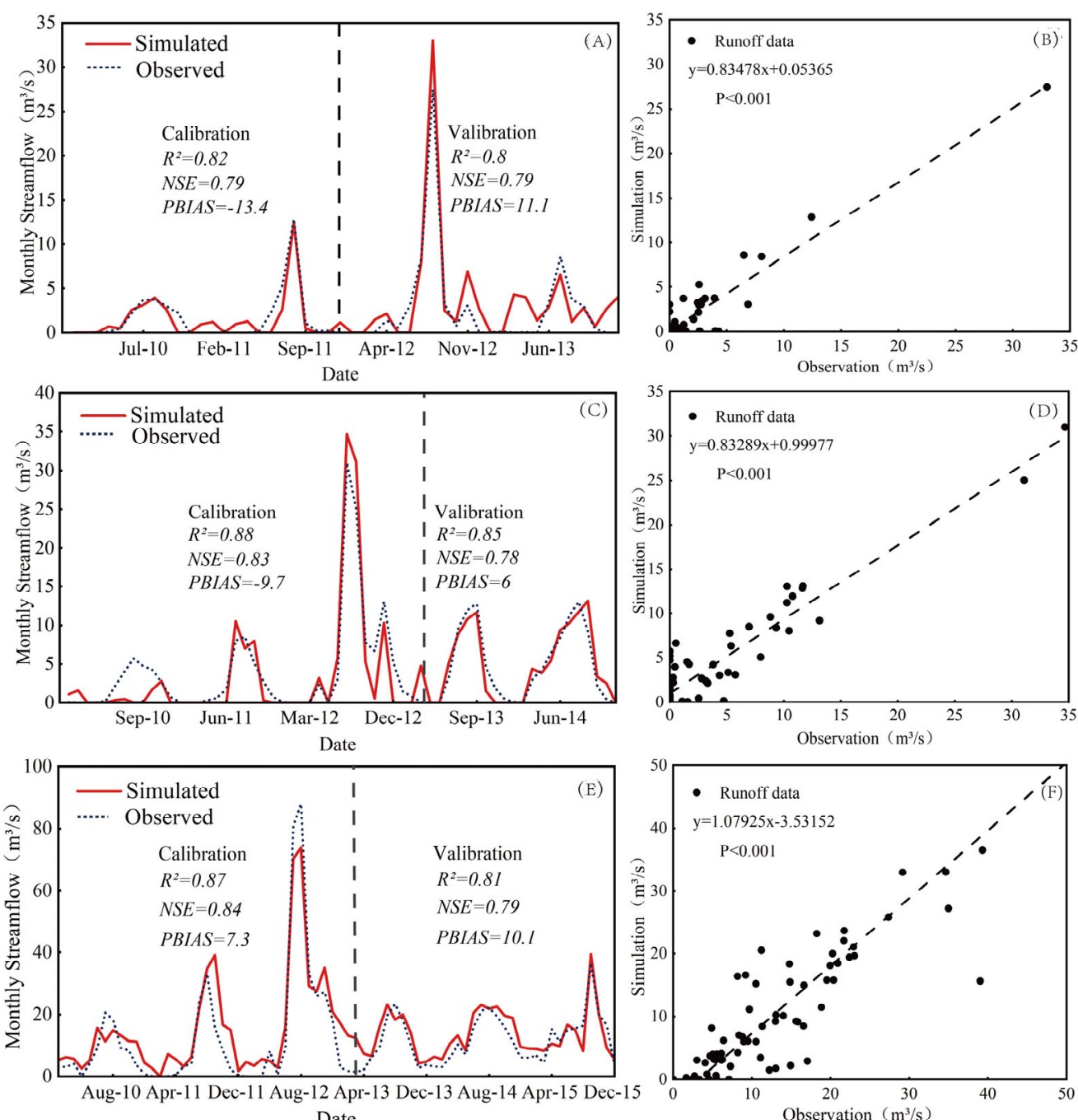

**Figure 5.** Hydrographs and scatter plots of the model calibration and validation for Jiuwangzhuang (**A**,**B**), Erdaozha (**C**,**D**), and Haihezha (**E**,**F**) hydrological stations.

The measured values of the 2012 flood peak are low. A significant error compared with the simulated values is probably related to the heavy rainfall in the study area in 2012. When the monthly runoff had low values during the verification period, it was found that the simulated runoff of the Jiyun River was higher than the measured average runoff. The average flow difference between the monthly simulated runoff and measured runoff in the verification period was 0.76 $m^3 \cdot s^{-1}$. The 2012 flood peak simulated values are high, and a low base flow is shown in the simulated results for the four years from 2010 to 2013. There is little snowfall in the study area, and rainfall mainly recharges surface water. From November to April, the rain is low, and the corresponding surface water volume passing

into the river is also low. This is another reason for the low simulated values during the non-flood period.

The calibration period models $R^2$, *NSE*, and *PBIAS* were 0.82, 0.79, and −13.4, respectively. The validation period models $R^2$, *NSE*, and *PBIAS* were 0.8, 0.79, and 11.1, respectively. Calibration and validation period coefficient $R^2 > 0.80$, *NSE* > 0.70, and |*PBIAS*| < 25, indicating that the simulation results have reached the standard of good or close to good, and the results are acceptable. The above analysis shows that the SWAT model constructed for the Jiyun River basin can simulate local hydrological processes. This indicates that the CMADS dataset is a reasonable basis for making the SWAT model and can be used for runoff simulation in the Jiyun River basin.

### 3.3. Analysis of Simulation Scenarios with Different Thresholds

To resolve the sub-basin delineation problem, we investigated the influence of the catchment area threshold on the control of sub-basin size and model simulation results. This study selected the nine sensitivity parameters mentioned above as the model rate parameters. SWAT models with different thresholds were developed. The results of the twelve model scenarios and the evaluation coefficients of the simulation results relative to the measured results are shown in Table 5.

**Table 5.** Monthly scale simulation evaluation results of different catchment area thresholds.

| Scheme | Catchment Area Threshold/km$^2$ | Proportion of Area | Number of Sub-Basins | Calibration | | | Validation | | | Annual Runoff |
|---|---|---|---|---|---|---|---|---|---|---|
| | | | | $R^2$ | *NSE* | *PBIAS* | $R^2$ | *NSE* | *PBIAS* | |
| 1 | 17 | 0.52% | 1022 | 0.46 | 0.45 | −2.1 | 0.75 | 0.73 | 14.8 | 27.83 |
| 2 | 27 | 0.82% | 616 | 0.61 | 0.6 | −2.7 | 0.77 | 0.76 | 14.3 | 27.42 |
| 3 | 37 | 1.12% | 465 | 0.7 | 0.7 | −4.8 | 0.8 | 0.8 | 12.3 | 26.74 |
| 4 | 47 | 1.42% | 359 | 0.79 | 0.78 | −4.7 | 0.84 | 0.83 | 13.2 | 26.22 |
| 5 | 57 | 1.73% | 315 | 0.82 | 0.79 | −13.4 | 0.8 | 0.79 | 11.1 | 26.21 |
| 6 | 67 | 2.03% | 277 | 0.82 | 0.79 | −13.6 | 0.79 | 0.78 | 11.1 | 26.2 |
| 7 | 77 | 2.33% | 241 | 0.81 | 0.8 | −7.2 | 0.77 | 0.76 | 16.5 | 26.19 |
| 8 | 87 | 2.64% | 215 | 0.81 | 0.8 | −7.3 | 0.75 | 0.74 | 16.5 | 26.4 |
| 9 | 97 | 2.94% | 201 | 0.82 | 0.82 | −7.4 | 0.73 | 0.72 | 16.4 | 26.52 |
| 10 | 107 | 3.24% | 183 | 0.81 | 0.8 | −4.9 | 0.74 | 0.72 | 19 | 26.42 |
| 11 | 117 | 3.55% | 163 | 0.81 | 0.81 | −4.6 | 0.7 | 0.69 | 19.5 | 26.11 |
| 12 | 147 | 4.45% | 143 | 0.79 | 0.78 | −4.8 | 0.65 | 0.64 | 19.5 | 25.87 |

As sub-basins increase, the modeled mean annual runoff decreases and becomes relatively stable. When the number of sub-basin divisions changes from 17 to 47, the average yearly runoff value decreases more; when the number of sub-basin divisions is between 47 and 87, the average annual runoff value tends to stabilize; when the number of sub-basin divisions exceeds 85, the average yearly runoff value tends to decrease slightly with the increase in the number of sub-basin divisions, but also tends to stabilize, with values ranging from 26.19 to 26.22 m$^3 \cdot$s$^{-1}$. The values fluctuate within a small range. It can be seen that there is a level of sub-basin division in the Jiyun River basin that stabilizes the basin runoff simulation. When below this level, the number of divisions influences the simulation results.

From the simulation evaluation results of the SWAT model of the Jiyun River basin, we can observe that: the $R^2$ of Scenarios 5, 6, and 9 were all as high as 0.82 in the calibration period, with the best correlation between the simulation results and the measured results; the best match of the simulation effect was Scenario 9, with an *NSE* of 0.82; the $R^2$ of Scenario 1 was 0.61, and the *NSE* was 0.60, which was the worst degree of correlation and match of the simulation effect. In the validation period, Scenario 4 showed a good correlation between the simulation results and the measured results and good agreement of the simulation effect, with an $R^2$ of up to 0.84 and *NSE* of up to 0.83. The worst degree of

correlation and agreement of the simulation effect was in Scenario 12, where the $R^2$ was 0.65 and the *NSE* was 0.64. The percentage deviations of all 12 scenarios were within 25%.

By utilizing the power function tangent to a straight line, the inflection point between the rate of change in river network density and the catchment threshold can be mathematically derived in order to identify the optimal threshold value of the basin. The simulation results were found to be more accurate when the watershed area was between 1.42% and 2.33% of the total watershed area. To validate the results, a SWAT model was employed to simulate the runoff in the mainstream area of the Haihe River. The central stream system of the Haihe River has a catchment area of 2066 km$^2$, stretching from Sanchakou to Dagukou and featuring two hydrological stations, Erdao Gate and Haihe Gate. The relationship between river network density and catchment area threshold yields a river network density function of $y = 112.166x^{-0.3572}$ and a threshold value of 40.5598 km$^2$. The results of the runoff simulation are illustrated in Figure 5C–F. The simulation results correspond to the observed values, where the determination coefficient $R^2$ and Nash–Sutcliffe coefficient *NSE* are 0.88 and 0.83 for the Erdao gate monitoring station and 0.74 and 0.71 for both in the validation period. At the Haihe Gate monitoring station, the rate periodic evaluation index had a determination coefficient $R^2$ of 0.87 and a Nash–Sutcliffe coefficient *NSE* of 0.84 during the determination period, which fulfilled the model's evaluation criteria. The $R^2$ and *NSE* were 0.77 and 0.76 during the validation period, respectively. The validation results of the SWAT model for the Haihe mainstem region demonstrate that the optimal threshold value of the basin can be determined by exploring the correlation between the river network density and the catchment area threshold, thus making the simulation results of the SWAT model more precise.

The exponential function determined the "inflection point" as the optimal catchment area threshold. This point was used to extend the threshold range and analyze the model results for different threshold ranges. Different approaches to setting catchment area thresholds affect the simulation results of the SWAT model. The simulation results of Scenario 1 were less, mainly because the number of sub-basin divisions was too large, and more pseudo-channels were generated and calculated in the river confluence process. The catchment area at this point was 0.52% of the total watershed area. The reason for the poor simulation results of scheme 12 was the number of sub-basins was too small, and the river network water system did not match the actual situation, leading to an unrealistic simulation. The catchment area at this point was 4.45% of the total watershed area. The overall evaluation of the simulation results was better, and the values of the evaluation indicators fluctuated less for catchment area thresholds between 47 km$^2$ and 77 km$^2$, with values of $R^2$ and *NSE* that were greater than 0.75 in both the validation period and the rating period. The catchment area in this threshold range was 1.42% to 2.33% of the total watershed area. The analysis showed a reasonable range for the catchment area threshold, closely related to the entire watershed area. The threshold should not be too large or small. The catchment area threshold should be kept at 1.42% to 2.33% of the total catchment area threshold when sub-basins are divided in the Jiyun River basin.

## 4. Conclusions

In this study, using the CMADS dataset combined with basin-related spatial and attribute data, the runoff simulation and analysis of the basin were conducted by applying the SWAT model under different catchment area thresholds with the upstream basin of Jiuwangzhuang hydrological station as the study area, and the following conclusions were obtained.

(1)  The combination of the river network density method and SWAT method quickly locked the reasonable catchment area threshold of the SWAT model, excluded the subjectivity and arbitrariness of the traditional SWAT model to extract the river network of the basin, and finally determined the functional expressions of river network density function and catchment area threshold in the study area: $y = 926.782x^{-0.47717}$.

(2) Various types of databases were constructed for the Jiyun River basin by using CMADS dataset combined with basin-related spatial and attribute data. The monthly-scale runoff processes were accurately simulated. In the scenarios with optimal catchment area thresholds, the $R^2$ for both the calibration and validation periods of the model rate was greater than 0.78, and the *NSE* was greater than 0.75, which was generally satisfactory for the simulation.

(3) Different threshold schemes had a direct impact on the simulation results of the SWAT model. The model results under different catchment area threshold conditions had upper and lower limits of the applicable range of thresholds. The model's simulation results were less when the thresholds were set too large or too small.

(4) From the simulation results of the catchment area threshold sequence, it could be concluded that the simulation results obtained from the pre-selected optimal threshold were within the final determined suitable threshold; thus, the corresponding exponential function could be established by the river network density method to lock the optimal threshold of the catchment area.

The SWAT model has good applicability in the Jiyun River basin by using the CMADS dataset. The river network density method combined with the SWAT model effectively extracts digital river networks. It is an efficient and reasonable approach to studying the proper range of thresholds from the optimal threshold of the "inflection point." It helps provide a scientific basis for determining the catchment area threshold and realizing the runoff simulation in the study area.

**Author Contributions:** Conceptualization, J.S.; data curation, Z.L. and S.J.; formal analysis, Z.L.; funding acquisition, J.S.; investigation, Z.L. and R.G.; resources, J.S.; supervision, J.S.; writing—original draft, Z.L.; writing—review and editing, J.S. All authors have read and agreed to the published version of the manuscript.

**Funding:** This research was financially supported by National Key R&D Program of China (2019YFC1407800), National Nature Science Foundation of China grants (41876134 and 41676112), the Changjiang Scholar Program of Chinese Ministry of Education (T2014253) to Jun Sun, and State Key Laboratory of Biogeology and Environmental Geology, China University of Geosciences (No. GKZ22Y656).

**Institutional Review Board Statement:** Not applicable.

**Informed Consent Statement:** Not applicable.

**Data Availability Statement:** All data are available from the authors upon request.

**Acknowledgments:** We thank all the members of National Key R&D Program of Ecological Environment Monitoring and Assessment and Pollution Control Technology in Bohai Bay, China (2019YFC1407800).

**Conflicts of Interest:** The authors declare no conflict of interest.

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
