# Peer review of "Runoff Simulation under the Effects of the Modified Soil Water Assessment Tool (SWAT) Model in the Jiyun River Basin"

_water, doi:10.3390/w15112110_

Round 1

Reviewer 1 Report (New Reviewer)

The manuscript used the mathematical value method of power function tangent to a straight line to obtain the optimal threshold value for the basin. They performed a quality assessment of the CMADS datasets and their reliability and applicability for runoff simulation in the Jiyun River basin. Finally, the obtained optimal threshold value was applied to the mainstream area of Haihe River in Tianjin, and a good verification effect was achieved. Overall, this manuscript is reasonable in structure. Before publishing this paper, I put forward some minor issues for the author's consideration.

1. The author can use more recent studies to illustrate the application of hydrological models in hydrological basins in the first paragraph of the "Introduction" section.

2. Please cite relevant literature to demonstrate the point ofThere are differences in the fusion methods and data sources used in the generation process of different datasetsin the first paragraph of the "Introduction" section.

3. In the last paragraph of the "Sensitivity Analysis" Section, the author only explained the physical significance of the four parameters and did not explain the impact of other parameters on the model in detail.

4. PBIAS is percent bias. In Table 2, Replace "PBIAS" with "PBIAS(%)."

5. In the chapter on "Data Analysis," the author analyzed the composition and proportion of land use and soil in the study area but did not further explain the influence of different types of land use composition and soil on hydrology.

Author Response

Reviewer 2 Report (New Reviewer)

Title: Application of SWAT model with CMADS data and catchment area threshold for runoff simulation in the Jiyun River basin

The paper reports simulation of runoff simulation in the Jiyun River basin using the SWAT framework. However, there are many uncertainties due to the poor preparation. This may not be useful to other watersheds. Furthermore, the authors need to articulate more clearly what the novel contribution is regards their publication. My detailed comments are listed below:

1.     Title is unappreciated. In a title, any abbreviation should be avoided, such as “CMADS”. In SWAT, it is well-known that meteorological data is required in SWAT model. In addition, “catchment area threshold” is too general.  

2.     I have main concerns about Introduction section. The literature review is obviously incomplete. This paper used the SWAT to simulate runoff. However, it is unclear what have been done in runoff simulations using SWAT model. What problems of runoff modelling remains to be addressed in SWAT model?  The authors need to highlight the novelties.

1.     In the first paragraph of Introduction, the authors stated “ For some watersheds, the distribution of meteorological stations in the study area was sparse or non-existent, reducing the model's reliability and increasing uncertainty.” It is unclear if the stations in your studied watershed was sparse or non-existent. You should show their distribution. Furthermore, it is unclear why you used Fuka et al.’s model. Again, although there are different datasets and analyzed methods, you should category them. Then you could point out what problems of your datasets were. Finally, why did you select Fuka et al.’s model?

4.     In the second paragraph of Introduction, it is unclear what problems were for CMADS series dataset. There are inconsistences. While you mentioned that the CMADS performed well, why did you need Fuka et al.’s model? Normally, SWAT uses daily weather data. Some missed data, the SWAT can generate them automatically. No special treatment will be required for weather data.

5.     In the third paragraph of Introduction, you should define “catchment area thresholds” before you used it. It is unclear why “the value of the catchment area had a more significant impact on the simulation results than parameter adjustment”. Generally, for the SWAT model, its accuracy will depend on the spatial resolution rather than a watershed area. That is, the spatial resolution the SWAT is the HRUs.

6.     The problems in both “CMADS data” and “catchment area thresholds” have not been described well. Therefore, your objectives are unclear.

7.     In Methodology, what spatial resolutions are DEM and land uses?  You should provide websites of all data sources.

8.     For meteorological data, you may present it how many monitoring stations were in this watershed, and what time steps were. Please mention if any uncertainty or missing data.

9.     You should present the calibration and validation data. It is unclear that the streamflow data were used for calibration and validation while your title was about runoff. Why did you not used runoff data for the calibration and validation?

10.  While your title is on runoff, I have not seen the results of runoff simulation.

11.  You should move the sections of calibration and validation and sensitivity analysis to before the section of the results and discussion.  

12.  In Results and discussion, the results should include effects of different catchment area thresholds on the spatial distribution of streamflow and runoff and their seasonal variation.

Author Response

Reviewer 3 Report (New Reviewer)

1. In the literature cited (1-7) in the introduction there is only the use of the SWAT model. It is more appropriate to have articles of a more general nature for the use of different hydrological models.

2. The weakness of DEM information and DEM data resolution are commented on, but there is no information about the source of this DEM. (2.2. "Data Analysis"). It is important for further analyses,   the drainage density is under the influence of DEM resolution, the higher the resolution is, the larger the drainage density will be under the same threshold value

3. In 2.6. "Evaluation criteria of the model, R², characterizes the correlation between the measured variables", probably referring to the relationship between measured and simulated.

4. In the description of the study area and in Fig. 1. "Location of the study area and distribution of the hydrological and meteorological gauges" there is only one hydrological station but in the results are presented  three hydrometric stations.

5. Only the final model calibration results (hydrographs and scatterplots of the simulated vs observed discharges and the statistical estimates) are presented and not the process of model calibration. From the statement in item 3.3 " 3.3 Calibration and Validation" it is not clear what the calibration and validation periods are.  What is the warm-up period?

There are no details of model parametrization – are all of the selected parameters calibrated for each subbasin or some of the subbasins?; are any or all of the selected parameters calibrated for some HRUs or specific land use class or soil type? In 3.4. the authors say: "This study selected the nine sensitivity parameters mentioned above as the model rate parameters. SWAT models with different thresholds were developed". It is not clearly stated what rated parameters stand for – is it that these parameters were used to calibrate each of the 12 new developed SWAT models or these already calibrated parameters were applied to the 12 new SWAT models? If the latter then lacking information of the calibration process brings into question the so proposed parameter regionalization attempt. Each of these 12 new developed SWAT models has different number of subbasins which in turn means that each of these 12 models differ not only in shape and areas of the subbasins but also in the number of HRUs, HRU distribution and channel length within the subbasins. Thus flood routing differs for each of the 12 new developed subbasins, given the fact that in SWAT the hydrological response is simulated for each HRU within the subbasin before being aggregated to the subbasin.  That is why at this point it is found difficult to evaluate the reliability of the results presented nor the robustness of the methodology proposed in this paper.

6. Many papers and studies recommend that each of the models be calibrated before analyzing the impact of the threshold area for watershed delineation.

If however the selected sensitive parameters were used to calibrate each of the 12 new developed models then the results presented in this research do support the importance of the area threshold for the delineation of a watershed, especially for large-scale and complex watersheds.

Round 2

Reviewer 2 Report (New Reviewer)

The authors have addressed my comments. I would recommend it after minor revision.

When presenting authors of references, all initials should be deleted. Only surname is sufficient. 

This manuscript is a resubmission of an earlier submission. The following is a list of the peer review reports and author responses from that submission.

Round 1

Reviewer 1 Report

The current study provides the sub-basin delineation for SWAT in the Jiyun River basin. This is my second time reviewing this article. The manuscript quality has been improved relative to the previous version. However, the following are my main concerns:

a) The objective of the study should be clearly mentioned in the last paragraph of the “Introduction” section.

b) There are some phrases such as “domestic and foreign scholars” and “Most studies at home and abroad” which seem unsuitable for an academic text, despite they may be grammatically correct. Please go through the text and replace such phrases with academic ones.

c) The study has been conducted in a limited area, indicating the fact that the delineation is site-specific. Therefore, I suggest mentioning that the study is site-specific, probably depending on the hydrological, geological, and climatic conditions, and such study should be carried out for the other regions.

d) Are the data, provided by the Spatio-Temporal Multiscale Analysis System, reanalysis products?

e) Replace “ percentage deviation (PBIAS)” with “percent bias (PBIAS)”.

f) Please denote that Moriasi et al., 2007 suggested the criterion of PBIAS > 25% for streamflow. They mentioned different criteria for different outputs, i.e. streamflow, sediment, contaminant load.

Reviewer 2 Report

This study determines a suitable catchment area threshold by combining the river network density method with the SWAT models selecting the Jiyun River watershed as a study area. If the reviewer remembers correctly, this manuscript was previously submitted under a different title and was returned once. The authors have then revised and resubmitted. The reviewer's opinion on the previous submission was that the threshold values obtained in this paper lacked a physical basis, that it was unclear whether they could be applied to different watersheds, and that the manuscript had little value. This shortcoming has not been resolved in any way, and the same opinion must be expressed here again. The reviewer recommends that the reviewers return the manuscript.
